# Synthetic Transformations and Medicinal Significance of 1,2,3-Thiadiazoles Derivatives: An Update

Ali Irfan [1][ID], Sami Ullah [2], Ayesha Anum [3], Nazish Jabeen [2], Ameer Fawad Zahoor [1,*][ID], Hafza Kanwal [2], Katarzyna Kotwica-Mojzych [4] and Mariusz Mojzych [5,*]

1 Department of Chemistry, Faculty of Physical Sciences, Government College University Faisalabad, Faisalabad 38040, Pakistan; raialiirfan@gmail.com
2 Department of Chemistry, Faculty of Sciences, Sargodha Campus, The University of Lahore, Sargodha 40100, Pakistan; sami.ullah@sgd.uol.edu.pk (S.U.); nazishjabeen1982@gmail.com (N.J.); kanwahfza@gmail.com (H.K.)
3 Islamabad Campus, Hamdard University of Pharmaceutical Sciences, Islamabad 44000, Pakistan; ayesha.anum@hamdard.edu
4 Department of Histology, Embryology and Cytophysiology, Medical University of Lublin, 20-080 Lublin, Poland; katarzynakotwicamojzych@umlub.pl
5 Department of Chemistry, Siedlce University of Natural Sciences and Humanities, 08-110 Siedlce, Poland
* Correspondence: fawad.zahoor@gcuf.edu.pk (A.F.Z.); mariusz.mojzych@uph.edu.pl (M.M.)

**Abstract:** The 1,2,3-thiadiazole moiety occupies a significant and prominent position among privileged heterocyclic templates in the field of medicine, pharmacology and pharmaceutics due to its broad spectrum of biological activities. The 1,2,3-thiadiazole hybrid structures showed myriad biomedical activities such as antifungal, antiviral, insecticidal, antiamoebic, anticancer and plant activators, etc. In the present review, various synthetic transformations and approaches are highlighted to furnish 1,2,3-thiadiazole scaffolds along with different pharmaceutical and pharmacological activities by virtue of the presence of the 1,2,3-thiadiazole framework on the basis of structure–activity relationship (SAR). The discussion in this review article will attract the attention of synthetic and medicinal researchers to explore 1,2,3-thiadiazole structural motifs for future therapeutic agents.

**Keywords:** 1,2,3-thiadiazole; synthetic strategies; biological activities; antiviral; anticancer; antifungal; insecticidal

## 1. Introduction

The fascinating aromatic 1,2,3-thiadiazole is a structurally active pharmacophore and great interest for researchers due to its versatile and wide array of biological activities in the field of medicine, pharmacology and pharmaceutics. Thiadiazoles occur naturally in four different isomeric forms, having one sulfur and two nitrogen atoms with hydrogen binding domain as presented in Figure 1 [1–4].

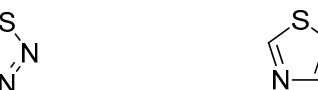

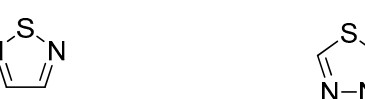

**Figure 1.** Structures of different isomeric forms of thiadiazole (**1–4**).

Thiadiazole scaffold emerges as an interesting structural motif due to the presence of different heteroatoms in its structure and its promising therapeutic activities in the treatment and cure of several diseases. Different substitutions on the thiadiazole nucleus improve the structure–activity relationship (SAR) of thiadiazole hybrid structures to enhance its therapeutic efficacy against a wide variety of pathogens [1,5]. Thiadiazole scaffolds exhibit a wide range of biological activities including antimicrobial [6–11], carbonic anhydrase inhibitors [12–14], antifungal [15–19], antibacterial [20–25], systemic acquired resistance [26,27], insecticidal activity [28,29], antiviral [30–33], antioxidant [34,35], anticonvulsant [36–42], antihypertensive activity [43,44], anticancer and antitumor [45–51], analgesic [52,53], anti-inflammatory [54–58], plant activator [59,60], antidiabetic [61], antileishmanial activity [62–65] and antituberculosis [66–71]. The different marketed drugs of thiadiazole moiety (**5**–**14**) are given below [26,72–76] (Figure 2).

**Figure 2.** Commercial drugs based on different thiadiazole scaffolds (**5**–**14**).

The bioactive 1,2,3-thiadiazole analogues display excellent and outstanding new profiles of medicinal, pharmacological and pharmaceutical activities [1,77]. In this review article, the most recent and relevant synthetic approaches and pharmacological activities of 1,2,3-thiadiazole analogues are discussed and can be classified into antiviral, insecticidal, antifungal and anticancer activities.

## 2. Synthetic Approaches

The different synthetic approaches and methodologies such as Hurd–Mori cyclization, ultrasound, microwave, multi-component reaction, metal-free conditions, [4 + 1] annulation, oxidative coupling intermolecular [3 + 2] heterocyclizations and heterogeneous catalysis are listed to furnish bioactive 1,2,3-thiadiazoles.

### 2.1. Hurd–Mori and Lalezari Cyclization

Pyrazolyl-1,2,3-thiadiazole scaffolds were afforded through carbon-sulfur, carbon double bonded with nitrogen and nitrogen-sulfur bonds formation by applying Hurd–Mori cyclization conditions as described in Scheme 1. In this efficient synthetic approach, various pyrazolyl-phenylethanones (**15**) were reacted with semicarbazide (**16**) to generate corresponding semicarbazone (**17**) intermediate that cyclized into substituted 1,2,3-thiadiazoles (**18**) after treating with thionyl chloride in good to excellent yield (Scheme 1) [72].

**Scheme 1.** Construction of pyrazole-based thiadiazoles **18**.

A similar one-pot simple synthetic strategy is presented in Scheme 2. Ketones having alkyl and aryl ring substituents (**19**) were treated with semicarbazide to afford appropriate semicarbazones (**21**), which underwent cyclization in excess of thionyl chloride via the Hurd-Mori process to achieve 1,2,3-thiadiazole hybrids (**21**) [78].

**Scheme 2.** Synthesis of 1,2,3-thiadiazoles **21** from semicarbazone.

In another example, semicarbazide hydrochloride was reacted with 2-oxoallobetulin (**22**) to furnish semicarbazone (**23**) intermediate which cyclized into thiadiazole (**24**) using SOCl2 via Hurd-Mori protocol (Scheme 3). The regioselectivity of the functionalization was mostly centered at the C-3 position for the product [79].

**Scheme 3.** Synthesis of 1,2,3-thiadiazole derivatives from 2-oxoallobetulin.

Kumar et al. reported a simple and easy approach for the synthesis of thiadiazole derivatives, accomplished under mild reaction conditions. In their pathway, ionic liquids

sulfonyl hydrazine (**25**) reacted with ketones or diketones (**26**) to provide ionic liquid hydrazone (**27**) that further treated with SOCl2 furnished substituted 1,2,3-thiadiazoles (**28**) in excellent (80–91%) yield (Scheme 4) [80].

R= $C_6H_5$, 4-$OCH_3C_6H_4$, 4-$CH_3C_6H_4$, $C_{10}H_7$, 4-$FC_6H_4$, 4-$ClC_6H_4$, 4-$BrC_6H_4$, 4-$NO_2C_6H_4$, $C_5H_4N$, $C_4H_3S$

**Scheme 4.** Synthesis of 1,2,3-thiadiazoles via ionic liquid support.

A facile, practical and improved Hurd–Mori approach to obtain 1,2,3-thiadiazoles in 44–98% yield range was presented by Chen et al. (Scheme 5). In this metal-free methodology, N-tosylhydrazones (**29**) and sulfur (**30**) were reacted in the presence of TBAI as a catalyst to achieve substituted aryl 1,2,3-thiadiazoles (31) [81].

R = -$CH_3$, -OCH3, F, Br, -CF3, Aryl, heterocyclic aryl, -CN, Cl

**Scheme 5.** Reaction of N-tosylhydrazones (**29**) with elemental sulfur for the synthesis of thiadiazole (**31**).

*2.2. Microwave-Assisted Synthesis of 1,2,3-Thiadiazoles*

Sun et al. described a multi-step and microwave-assisted technique in which hydrazine and the diethyl carbonate (**32**) were combine to achieve hydrazide intermediate (**33**) which treated with ethyl 3-oxobutanoate (**34**) produced suitable Schiff base derivative (**35**). The substituted 1,2,3-thiadiazole-5-carboxylate scaffold (**36**) was provided by cyclization of SOCl$_2$ with derivative (35) via a Hurd–Mori reaction. In the next step, the reaction of compound (**36**) with $N_2H_4 \cdot H_2O$ provided 1,2,3-thiadiazole acetanilide derivative (**37**) which was combined with substituted isothiocyanate to afford a substituted isothiocyanic ester (**38**). In the alkaline conditions, isothiocyanic ester (**38**) cyclized to 1,2,4-triazole with 1,2,3-thiadiazole substituent (**39**). The SH moiety on triazole was reacted with different alkyl or benzyl chlorides under microwave conditions to obtain the final products (**40**) in high yields within 15 min at 90 °C (Scheme 6) [82].

*2.3. Multi Component Synthetic Strategies*

Zheng et al. developed a simple, convenient and one-step multi-component synthetic approach of 1,2,3-thiadiazole via Ugi four-component reaction (U-4CR) (Scheme 7). The first step of this strategy was the generation of imine intermediate by mixing an amine (**42**) derivative with aldehyde (**43**). In the second step, imine intermediate was treated with thiadiazole (**41**) and isocyanide component (**44**) to obtain 5-methyl substituted thiadiazole derivatives (**45**) in low to excellent (6–98%) yield [83].

**Scheme 6.** Synthesis of triazole-based 1,2,3-thiadiazole derivatives (**49**).

X = 3-Cl,4-CH$_3$C$_6$H$_3$, 3-F,4-CH$_3$C$_6$H$_3$
Y = C$_6$H$_5$, 4-NO$_2$C$_6$H$_4$, 3-ClC$_6$H$_4$, 2-ClC$_6$H$_4$,
    4-FC$_6$H$_4$, 3-FC$_6$H$_4$, 4-ClC$_6$H$_4$, 4-CF$_3$C$_6$H$_4$,
    3-CF$_3$C$_6$H$_4$, 2-CH$_3$C$_6$H$_4$, 2-CF$_3$C$_6$H$_4$, 4-CH$_3$C$_6$H$_4$,
    3-CH$_3$C$_6$H$_4$, 3-NO$_2$C$_6$H$_4$, 4-OHC$_6$H$_4$, 3-OHC$_6$H$_4$
Z = Isopropyl, cyclohexyl

**Scheme 7.** 5-methyl substituted thiadiazole derivatives (**45**) synthesis via U-4CR.

Wang et al. utilized sodium borohydride for the reduction of the 5-carboxylate group of thiadiazole (**46**) to corresponding alcohol (**47**) which readily combined with pyridinium chlorochromate (PCC) to achieve 5-carbaldehyde substituted thiadiazole (**48**). The substituted amines, cyclohexyl isocyanide and azidotrimethylsilane were treated with (**48**) scaffold in methanol via Ugi-4CR to obtain tetrazole moiety containing 4-methyl substituted 1,2,3-thiadiazoles (**49**) in moderate yield (Scheme 8) [84].

The reaction of methyl ketones (**50**) with *p*-toluenesulfonyl hydrazide (**51**) and KSCN in DMSO solvent afforded corresponding thiadiazoles (**53**) as reported by Wang et al. This reaction was facilitated by I2 and CuCl$_2$, resultantly obtained 71–89% yield range in case of aryl-substituted thiadiazoles while 48–74% yield range was reserved for alkyl-substituted thiadiazoles (Scheme 9) [85].



**Scheme 8.** Preparation of tetrazole ring containing thiadiazole derivatives (**49**).

**Scheme 9.** I$_2$/CuCl$_2$ mediated catalysis for the construction of thiadiazoles (**53**).

### 2.4. Ultrasonic Assisted Synthesis of 1,2,3-Thiadiazoles

An interesting example of the use of the ultrasonic-assisted technique is the synthesis described by Dong et al. (Scheme 10). In this multi-step synthetic route, the 5-carboxylic acid substituted thiadiazole (**54**) was reacted with thionyl chloride to obtain its corresponding 5-carbonyl chloride derivative (**55**), which in the next step was combined with glycine to afford carboxamide derivative (**56**). The last one (**56**) was reacted with substituted benzaldehyde and Ac$_2$O under the ultrasonic-assisted irradiations conditions to furnish 4-benzylideneoxazole moiety containing 4-methyl-1,2,3-thiadiazole (**57**). In further steps, the oxazole derivative (**57**) was subjected to reaction with piperidine to obtain the intermediate (**58**) which further halogenated in chloroform to afford final substituted phenyl acrylamide 1,2,3-thiadiazole derivatives (**59**) in 73.9–99% yield [86].

### 2.5. Multi-Step Synthesis of 1,2,3-Thiadiazole from Quinolin-8-ol

Hayat et al. reported the synthetic route to achieve substituted 1,2,3-thiadiazole derivatives and investigated their pharmacological profile (Scheme 11). Ethyl substituted quinolineoxy acetate (**61**) was afforded by treating ethyl chloroacetate with quinolin-8-ol (**60**) in refluxing conditions which upon reaction with N$_2$H$_4$·H$_2$O furnished 2-(quinolin-8-yloxy)aceto hydrazide (**62**). Next, the condensation reaction between substituted aromatic aldehydes and (**62**) to afford quinolinloxy acetohydrazone (**63**), which in the presence of SOCl$_2$ cyclized into targeted thiadiazole derivatives (**64**) [87].

**Scheme 10.** Synthesis of acrylamide derivatives of thiadiazoles (**59**).

**Scheme 11.** Acetohydrazone as starting precursor for the preparation of thiadiazoles (**64**).

### 2.6. Hydrolization and Esterification Approach

The simple synthetic route for carboxylate derivatives of thiadiazoles (**67**) is outlined in Scheme 12 and include two steps, i.e., first hydrolysis of thiadiazole (**65**) to 6-carboxylic acid derivative of thiadiazole (**66**), followed by esterification with various substituted alcohols afforded desired derivatives (**67**) [88].

R = CH₂CH₃, CH₂CF₂CF₃, CH₂CH₂CH₂CH₂CH₃, CH₂CF₃, CH(CF₃)₂, CH₂CF₂CF₂CF₂CHF₂, CH₂CH₂CH₃, CH₂CH₂CH₂CH₃, CH₂CH(OH)CH₂OH, CH(CH₃)₂, CH₂CF₂CF₂CF₃, CH₂C₆F₅

**Scheme 12.** Synthesis of new carboxylate derivatives of thiadiazoles (**67**).

### 2.7. Transition Metal-Free Synthetic Approach

The 5-acyl-1,2,3-thiadiazoles scaffolds (**71**) were afforded by I2/DMSO-mediated cross-coupling reaction of three components, i.e., enaminones (**68**), elemental sulfur (**69**) and tosylhydrazine (**70**) (Scheme 13). Broad functional groups tolerance with targeted compounds up to 92% yield are the advantages of this methodology [89].

**Scheme 13.** Synthesis of substituted 1,2,3-thiadiazole (**71**) via cross-coupling mediated by I₂/DMSO.

### 2.8. Nucleophilic Addition

A new simple operational and inexpensive strategy was developed to construct thiadiazole motifs (**75**) by the reaction of α-diazo carbonyl compounds (**72**) with BnBr (**74**) and CS₂ (**73**) as depicted in Scheme 14 [90].

**Scheme 14.** Synthesis of 1,2,3-thiadiazoles (**75**) via nucleophilic addition.

### 2.9. [4 + 1]. Annulation of Azoalkenes

A new sustainable, regioselective, environmentally friendly and broad functional-group compatibility synthetic strategy for thiadiazoles was described by Zhang et al. In their protocol, photocatalysis reaction of azoalkenes (**76**) with KSCN in the presence of cercosporin and ᵗBuOK afforded desired thiadiazoles (**77**). Electron donating groups showed maximum yield, week EWD groups displayed good yield. Strong EWD groups afforded moderate yield, heterocyclic moiety such as furan showed the least yield while pyridine did not furnish yield under the conditions as mentioned in Scheme 15 [91].

**Scheme 15.** Photocatalysis reaction of azoalkenes (**76**) with KSCN to afford 1,2,3-thiadiazoles (**77**).

### 2.10. Intramolecular Oxidative Nucleophilic Substitution of Hydrogen (ONSH)

In this basic media strategy in Scheme 16, thiadiazoles (**78**) with $I_2$ as an oxidant in the presence of 1.1 equivalent of $^tBuOK$ provided 1,2,3-thiadiazoles (**79**) [92].

R = NO$_2$, NHAc

88–91% Excellent yield

**Scheme 16.** Preparation of fused 1,2,3-thiadiazoles via ONSH mechanism.

### 2.11. Intermolecular [3 + 2] Heterocyclization

Fan et al. presented a metal-free, eco-friendly, flexible structural modifications and mild reaction conditions methodology in which combination of $I_2$ and $O_2$ promoted inter-molecular [3 + 2] heterocyclization between aryl hydrazines (**80**) and triethylammonium thiolates (**81**) furnished 1,2,3-thiadiazle derivatives (**82**) up to 92% yield (Scheme 17) [93].

**Scheme 17.** $I_2/O_2$ mediated synthesis of 1,2,3-thiadiazoles (**82**).

### 2.12. Cornforth Rearrangement Approach

In this domino type reaction, *N*-heteroarylamidines based thiadiazole analogues (**85**) were furnished in moderate to good (58–76%) yield by treating acetamides (**83**) with imidazole (**84**) in EtOH/EtONa mixture (Scheme 18). This domino reaction mechanism is analogous to Cornforth rearrangement [94].

**Scheme 18.** *N*-Heteroarylamidines based thiadiazole analogues (**85**).

### 2.13. Cascade Process

Liu et al. reported a simple and facile synthetic route for thiadiazoles involving reaction of chlorinated ketones (**86**) with tosylhydrazine (**70**) to obtain intermediate (**87**) which further cyclized with sulfur reagent in the presence of basic medium to yield (38–68%) disubstituted 1,2,3-thiadiazoles (**88**). Bromophenyl displayed the maximum 68% yield in substrate scope, methoxycarbonyl phenyl afforded least 38% yield, while dihyroxyphenyl and nitrophenyl failed to give desired product in 0.0% yield (Scheme 19) [95].

R₁ = Phenyl, methyl phenyl, methoxyphenyl, trimethylphenyl, fluorophenyl chlorophenyl, bromophenyl,methoxycarbonylphenyl, difluorophenyl, dihydroxy phenyl, nitrophenyl, R₂ = H

**Scheme 19.** Synthesis of 1,2,3-thiadiazoles from in situ generated azoalkenes with $S_3^{\bullet-}$ via a cascade process.

## 3. Biological Activities of 1,2,3-Thiadiazole Derivatives

The main objective of the present section is the search and presentation of the 1,2,3-thiadiazole based most potent therapeutic agents for a variety of biological and pharmacological applications with little or no side effects. In this section, 1,2,3-thiadiazole derivatives have been divided into classes according to their biological activity.

### 3.1. Antiviral Agents

Pawar et al. described antiviral drugs as such bioactive compounds which are used for the treatment of viral infections [96]. Rossignol et al. reported that most of the antiviral agents are specifically targeted oriented while some other antivirals have a broad spectrum against a vast variety of viral infections [97]. Some antiviral drugs such as ribavirin (**89;** Figure 3) are being used against both RNA and DNA viruses [98–100], efavirenz (**90;** Figure 3) used for the treatment of HIV/AIDS [101–103] and arbidol (**91;** Figure 3) was used against influenza viruses [104,105].

**Figure 3.** Structures of antiviral drugs **89**–**91**.

Zhan et al. prepared 1,2,3-thiadiazole thioacetanilide by adopting a multi-step methodology and exhibited their anti-HIV (Human Immunodeficiency Virus) activity against MT-4 cells. Among all synthesized derivatives, the thioacetanilide based 1,2,3-thiadiazole scaffold (**92**; Figure 4) showed remarkably the best anti-HIV activity in terms of $EC_{50}$ value $0.059 \pm 0.02$ μM, $CC_{50} > 283.25$ μM, SI > 4883 μM compared to the reference compounds such as NVP (nevirapine) DLV (delaviridine), EFV (Efavirenz), AZT (azidothymidine) and VRX-480773. SAR displayed that the significant enhancement in the antiviral efficacy of compound **92** was due to the introduction of the nitro group on the phenyl ring. Anilide scaffold having $NO_2$ and halogen-substituted aryl ring at *o*-position took great part to enhance the anti-HIV potential, however, the case was the difference for fluorine atom which decreased activity at a certain level [106].

**Figure 4.** Thioacetanilide based 1,2,3-thiadiazole scaffold **92** exhibiting maximum anti-HIV potential.

Zhan et al. prepared potent 1,2,3-thiadiazole derivatives against HIV-1. The new 1,2,3-thiadiazole (**93**; Figure 5) proved as the most active anti HIV-1 agent ($EC_{50}$ value $0.0364 \pm 0.0038$ μM, $CC_{50} > 240.08$ μM and SI > 6460 μM) against MT-4 cells among the synthesized derivatives in comparison with the reference compounds NVP ($EC_{50}$ 0.208 μM) and DLV ($EC_{50}$ 0.320 μM) EFV ($EC_{50}$ 0.00440 μM) and AZT ($EC_{50}$ 0.0151 μM). Phenyl ring substituted with $2,4\text{-Br}_2$ group significantly increased the antiviral potential of compound **93** than all other substituents on the phenyl ring. SAR studies indicated decreasing order of antiviral strength, i.e., $2,4\text{-Br}_2 > 2,4\text{-Cl}_2 > 2,4\text{-F}_2$ [107].

Dong et al. prepared potent antiviral piperidine-based thiadiazole derivatives, and the SAR study showed that chlorine atom substituted compounds exhibited good antiviral activity and the substitution on phenyl ring affects inhibitory action. 1,2,3-thiadiazole (**94**; Figure 6) displayed excellent potency with $IC_{50}$ 3.59 μg/mL as compared to lamivudine ($IC_{50}$ value 14.8 μg/mL). It was observed that compounds having *p*-substituents exhibited more cytotoxic potency than the *o*-substituted derivatives. Moreover, lowering of the selective index with higher anti-HBV potencies of the synthesized compounds was observed when comparing the results with lamivudine. Compound **94** proved to be the most active one as compared to the other derivatives [86].

**Figure 5.** Anti-HIV activity of 5-(2,4-dibromophenyl)-1,2,3-thiadiazole **93**.

**Figure 6.** Antiviral activity of 1,2,3-thiadiazole **94**.

Synthesis and screening of antiviral potency of tetrazole-based 1,2,3-thiadiazoles structural against tobacco mosaic virus (TMV, anti-TMV) were evaluated [84]. The studies indicated that some obtained compounds (**95–98**; Figure 7) have higher anti-TMV activity (33.75–48.73%) than ribavirin at 100mg/mL concentration (33.23%) and comparable potency to ribavirin at 100 μg/mL. Structure **97** displayed the best protection effect among the synthesized derivatives as well as from the ribavirin—reference drug. Anti-TMV activity of synthesized scaffolds and reference drugs decreases in order of **98** > ninamycin >**97** > **95** > **96** > ribavirin while the protection effect decreased in order of ninamycin > **97** > **96** > **95**> ribavirin > **98**. Compound **98** display a higher inhibition potential, i.e., 48.73% as compared to ninamycin and ribavirin. The SAR study showed that the **2-fluorophenyl substituent** derivative **97** displayed **the best protection effect** among the synthesized congeners as well as from the ribavirin reference drug. The protection effect decreased in order of **ninamycin > 2-fluorophenyl > cyclopropyl > isopropyl > ribavirin > 4-ethylphenyl.** Overall the scaffold **97** proved to be a promising therapeutic agent, which has a better protection effect as well as the best anti-TMV activity [84].

**Figure 7.** Antiviral activity of terazole derivatives **95–98**.

Mao et al. reported the synthetic protocol to afford substituted methyl carbohydrazide-based 1,2,3-thiadiazoles and determined the anti-TMV activity. The thiophene containing

carbohydrazide 1,2,3-thiadiazole (**99**; Figure 8) showed potent direct anti-TMV activity with 58.72% and 61.03% induction potencies at 50 µg/mL when compared with reference drugs, ninamycin and tiadinil. Both the standard reference drugs, ninamycin and tiadinil, exhibited anti-TMV activity (54.93% and 7.94%) and induction activity (18.58% and 59.25%) respectively in vivo at 50 µg/mL concentration. The scaffold **99** showed remarkably and significant higher antiviral potential than the standard reference drugs. The excellent anti-TMV activity of analogue **99** was due to the presence of thiophene moiety as compare to all the other tested compounds. The structural motif **99** showed significantly higher induction activity (61.03%) more than the induction activities of reference drugs, ninamycin and tiadinil (18.58%) and (59.25%), respectively. So the derivative **99** proved to be a plant elicitor and anti-TMV agent simultaneously [108].

**99**

**Figure 8.** Antiviral 1,2,3-thiadiazole scaffold **99**.

Another example of compounds with anti-TMV activity are structures **100** and **101** with a 1,3,4-oxadiazole ring (Figure 9) [109]. They showed potent curative, inactivation and protection effects against TMV. The derivative **100** displayed curative rate 54.1%, inactivation rate 90.3% and protection rate 52.8% while the compound **101** showed 47.1, 85.5 and 46.4% curative, inactivation and protection rates, respectively. The ningnanmycin was used as a standard reference drug, exhibiting a 56.1% curative rate, 92.5% inactivation rate and 59.3% protection rate. The incorporation of 1,3,4-oxadiazole moiety in the skeleton of 1,2,3-thiadiazoles significantly increased the anti-TMV activity [109].

**100**　　　　　　　　　　　　　　　　　　**101**

**Figure 9.** 1,2,3-thiadiazoles **100** and **101** with anti-TMV activity.

Other promising anti-TMV agents **102** and **103** were presented by Zheng et al. (Figure 10) [83]. The substituted 1,2,3-thiadiazole-4-carboxamide scaffold **102** exhibited the best antiviral curative activity of 60 and 47%, at 500 and 100 µg/mL concentrations, while the standard tiadinil showed curative activity 58 and 46% at 500 and 100 µg/mL concentrations, respectively. Compound **103** showed its potent protective effect 76 and 71% at 500 and 100 µg/mL concentrations in comparison with the standard drug tiadinil, which displayed protective 75 and 57% at 500 and 100 µg/mL. The structure **103** marked a higher protective effect than the tiadinil and **102** at 100 µg/mL concentration. The enhanced protective potential of **103** was dose-dependent and due to the presence of hyroxyphenyl moiety. The fluorophenyl, chlorophenyl and hyroxyphenyl functionalities contributed to better efficacy that led to a future investigation to develop promising anti-TMV agents [83].

**Figure 10.** 1,2,3-Thiadiazole scaffolds **102** and **103** with anti-TMV activity.

1,2,3-Thiadiazole acetanilide hybrid structures prepared by Zhan et al. were screened for their antiviral activity against HIV-1 [110]. In the group of achieved derivatives structure (**104**; Figure 11) exhibited moderate antiviral activity ($EC_{50}$ value $0.95 \pm 0.33$ μM) against HIV in comparison with the reference standard drugs such as: NVP ($EC_{50}$ value 0.208 μM), DLV ($EC_{50}$ value 0.32 μM), EFV ($EC_{50}$ value 0.00440 μM) and zidovudine ($EC_{50}$ value 0.0151 μM). The scaffold **104** displayed remarkable high and excellent anti-HIV-1 activity than all standards. The reason for the highest anti-HIV-I activity of structural motif **104** among the synthesized congeners was due to the 4-acetyl substituted anilide phenyl ring of 1,2,3-thiadiazole [110].

**Figure 11.** Antiviral activity of 4-(3,4-dichlorophenyl)-1,2,3-thiadiazole scaffold **104**.

### 3.2. Anticancer Agents

Cancer has drawn the attention of today's medical science all over the world because it is the lethal, notably complex, most prominent and serious threat to human health. It forms the lump of uncontrolled growth cells called tumors or neoplasm tumor cells. The neoplasm malignant are diversified heterogeneous cells which have properties of rapid proliferation and capability to invade or spread to other parts of the body through the bloodstream and lymphatic system. Most researchers have devoted their extensive research to developing effective anticancer agents, along with the employment and application of integrated surgical procedures, radiation therapy and chemotherapy [111,112]. Great advances and many discoveries in the development of a large quantity of anticancer therapeutics such as carboplatin (**105**) cisplatin (**106**) [113–115], anastrozole (**107**) [116,117] and doxorubicin (**108**) (Figure 12), etc., have been made over the past 60 years [118–121].

Noteworthy are the 4,5-diaryl-1,2,3-thiadiazole derivatives obtained by Wu et al. due to their method of synthesis and anticancer properties [122], and results pointed out that compounds **109** and **110** are the lead compounds by exhibiting favorable activity (Figure 13). Growth inhibition rate of tumor cells (obtained from mice S180) by derivative **109** at a dose of 40 mg/kg administrated for 5 days reached 81.0%, while compound **110** displayed a tumor suppression efficacy of 64.2% (at 100 mg/kg of dose administered to mice for five days). These effects were compared with the standard reference CA-4 (combretastatin A-4), which showed a 64.2% antitumor effect at a dose of 40 mg/kg used for

4 days in a mouse model. The SAR study indicated that the reason of the highest antitumor activity of structural motif **109** than reference drug CA-4 and derivative **110** was because of OMe group at C-4 and H at C-3 of 5-phenyl ring of 1,2,3-thiadiazole, while the replacement of hydrogen at C-3 with the nitro group dropped the antitumor activity of derivative **110**. The scaffold **109** could be a promising candidate due to its lower cytotoxicity and excellent antitumor effect [122].

**Figure 12.** Clinical anticancer drugs **105**–**108**.

**Figure 13.** Antitumor effect of 1,2,3-thiadiales **109** and **110**.

New 1,2,3-thiadiazole derivatives (**111** and **112**; Figure 14) obtained by Hosny et al. via Hurd–Mori cyclization were tested against MCF-7 tumor cells. The IC$_{50}$ values of compounds **111** and **112** (IC$_{50}$ = 12.8 µg/mL and 8.1 µg/mL, respectively) were comparable to doxorubicin as the standard anticancer drug with IC$_{50}$ = 3.13 µg/mL. The derivative **111** showed the highest anti-breast cancer activity than the scaffold **112** and reference drug doxorubicin [123].

**Figure 14.** Anticancer activity of 2-thioamide-1,2,3-thiadiazole **111** and **112**.

Cikotiene et al. reported the synthetic protocol to achieve the substituted 4,5-diaryl-1,2,3-thiadiazoles as Hsp90 chaperone protein inhibitors. This protein plays a significant part in developing tumor cells, so its inhibition is the main target of many anticancer drugs. The Hsp90 chaperone stabilizes a number of proteins that are essential for the growth of tumors, due to which researchers all over the world are interested in the development of Hsp90 inhibitors which would be investigated as anticancer agents. In the group of tested

derivatives, structure **113** deserves special attention as an inhibitor of cancer cells with $GI_{50}$ value of 0.69 μM for U2OS (osteosarcoma) cells and 0.70 μM for HeLa (cervical carcinoma) cells (Figure 15) [124]. Isothermal titration calorimetry was used to determine the binding affinity of compound (113) to Hsp90F and Hsp90N. The derivative (**39**) acted as a binder to both Hsp90N andHsp90F with the observed $K_d$ of about 42 nM and 37 nM, while for the reference compound17-AAG [17-(allylamino)-17-demethoxygeldanamycin] $K_d$ was 200 nM and 240 nM, respectively [124].

**Figure 15.** Anticancer activity of 1,2,3-thiadiazole derivative **113** against U2OS and HeLa cell lines.

Synthesized by Cui et al., dehydroepiandrosterone derivatives of thiadiazoles were investigated against T47D and HAF cell lines for their antitumor activity applying the sulforhodamine B (SRB) assay. In this group of compounds, derivative **114** (Figure 16) focuses particular attention, due to its strong antitumor and antimetastatic activities, in vitro and in vivo. Its $IC_{50}$ value for T47D cells was 0.058 ± 0.016 μM and for HAF cells $IC_{50}$ = 21.1 ± 5.06 μM. Moreover, the selectivity of compound **114** (SI = 364) was 214 folds better than adriamycin (positive control) (ADM; SI = 1.7). The SAR study showed that the introduction of 1,2,3-thiadiazole and D-proline chemical entities in derivative **114** significantly enhanced antiproliferative activity than the parent DHEA. The SI of scaffold **114** was 214 folds better than SI of ADM standard drugs. The derivative **114** could be used as the promising and novel class of anticancer agent due to its low cytotoxicity, high selectivity index and excellent antitumor and antiproliferative activities (Figure 16) [125].

**Figure 16.** 1,2,3-Thiadiazole dehydroepiandrosterone derivative **114** as new antitumor agents.

### 3.3. Insecticidal Agents

Insecticides are the designed substances of either chemical or biological origin that used to regulate the insect behaviour, control insects, cause death, dysfunction and moving away [126,127]. Some of the insecticidal agents, such as tebufenozide (**115**) act as molting hormones primarily against caterpillar pests [128]; imidacloprid (**116**) is the most well-known broad spectrum insecticide, which used as an insect neurotoxin [129–131], and neuroactive pymetrozine (**117**) is a novel class of pyridineazomethine insecticide used for the control of aphids and whiteflies in crop fields (Figure 17) [132].

An interesting example of 1,2,3-thiadiazole derivatives with insecticidal properties are two groups of *N-tert*-butyl-*N,N'*-diacylhydrazines obtained by Wang et al. [29] and tested their activity against *Plutella xylostella* L. and *Culex pipiens pallens*. Derivative **118** (Figure 18) displayed remarkable and significantly the highest insecticidal potential (79% mortality at 200 μg/mL) against *Plutella xylostella* L., while **119** exhibited 68% mortality against the same insects at concentration 200 μg/mL. Insecticidal activity of reference agent, i.e.,

tebufenozide, was 40%. The scaffold **118** displayed excellent and the highest insecticidal potency, 79% more than the insecticidal potencies of reference drug tebufenozide (40%) and derivative **119** (68%). The EWD group chloro increased the insecticidal activity of scaffold **118**, while ED group methyl decreased the insecticidal potency of scaffold **119** [29].

**Figure 17.** Known insecticidal agents **115–117**.

**Figure 18.** Insecticidal activity of *N-tert*-butyl-*N,N′*-diacylhydrazines **118** and **119**.

Zhang et al. designed the (*E*)-β-farnesene based carboxamides of thiadiazoles and checked their aphicidal behavior against *Myzus persicae* [133]. The three compound **120–122** (Figure 19) exhibited LC$_{50}$ values of 33.4 μg/mL, 50.2 μg/mL and 61.8 μg/mL, respectively. The 1,2,3-thiadiazole carboxamide analogues showed significantly higher aphicidal activity than (*E*)-ß-farnesene, but lesser insecticidal activity than pymetrozine insecticide (LC$_{50}$ = 7.1 μg/mL). It was observed that fluoro or difluoro groups on phenyl moiety enhanced the aphicidal potential significantly in novel chemical entities of Eβf 1,2,3-thiadiazole **120** and **121** while the methyl substitution led to lower aphicidal activity of scaffold **122** [133].

**Figure 19.** Insecticide Eβf 1,2,3-thiadiazole carboxamides **120–112**.

Pyrazole oxime derivatives, as a new Fenpyroximate analogue bearing a thiadiazole moiety, were synthesized by Dai et al. and then their insecticidal, acaricidal and cytotoxic activities were examined. The best insecticidal properties possessed oximes **123** and **124** (Figure 20) against *Aphis craccivora* at 100 μg/mL with 90% of the mortality rate. The structure–activity relationship study revealed that due to the insertion of F- and Me-groups at *p*-position of phenoxy moiety attached to the pyrazole oxime **123** and **124** displayed good insecticidal activities but slightly lower than commercial drug imidacloprid [134].

**Figure 20.** Ethers of pyrazole oximes **123** and **124** with insecticidal properties.

Another interesting example of insecticidal compounds are hybrids of substituted triazole and 1,2,3-thiadizole scaffold that were investigated for their activity against *Aphis laburni* and TMV by Li et al. [135]. Preliminary bioassays indicated compounds **125** and **126** (Figure 21) as lead compounds by depicting enhanced activity against *Aphis laburni* at 100mg/mL (mortality $\geq$ 95%) while the derivative **127** represent the least active compound (due to $CF_3$ substituted triazole ring) that displayed only 25.25% mortality. The structure–activity study indicated that scaffolds with aromatic moieties attached to the carbonyl carbon of triazole ring have higher insecticidal activity than scaffolds with alkyl moieties [135].

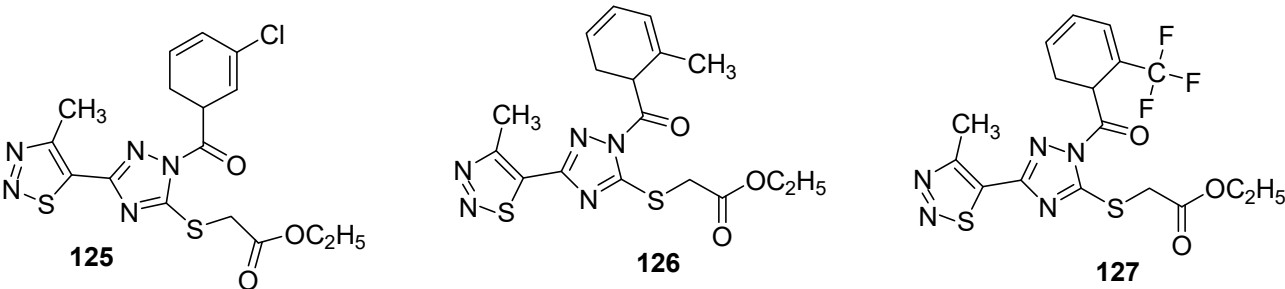

**Figure 21.** Substituted 4-methyl-1,2,3-thiadiazole derivatives with insecticidal properties against *Aphis laburni*.

### 3.4. Amoebicidal Agents

A parasitic disease caused by a protozoa *Entamoeba histolytica* is known as amoebiasis. *E. histolytica* has infected 40 to 50 million people worldwide and led to the development of amoebic colitis or extraintestinal abscess that has resulted in 100,000 deaths annually [136–138]. The tissue amoebiasis is treated by different drugs such as metronidazole, chloroquine, tinidazole, dehydroemetine and nitazoxanide, while the diloxanide furoate and iodoquinoline drugs are used to treat luminal infections (Figure 22) [139–143].

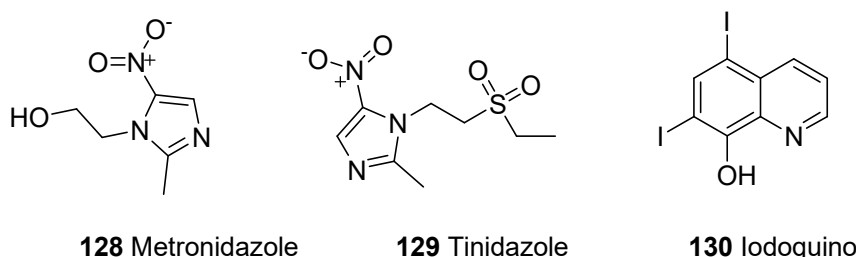

**128** Metronidazole     **129** Tinidazole     **130** Iodoquinol

**Figure 22.** Amoebicidal commercial dugs **128**–**130**.

Hayat et al. reported the synthetic route to achieve substituted 1,2,3-thiadiazole scaffolds from the cyclization of quinoline-based hydrazone. All substituted 1,2,3-thiadiazole hybrids and quinoline-based hydrazones were evaluated for antiamoebic activity against HM1 and IMSS strains of *E. histolytica*. The scaffold 4-bromo phenyl-1,2,3-thiadiazole (131, Figure 23) and furan based 1,2,3-thiadiazole (132, Figure 23) exhibited potent antiamoebic activity (IC50 = 0.24 μM and IC50 = 0.23 μM, respectively) when compared with metronidazole (IC50 = 1.80 μM). The parent intermediate quinoline-based hydrazone displayed better antiamoebic potential than the targeted derivatives of 1,2,3-thiadizoles. The SAR study revealed that the significant loss of antiamoebic potential in scaffolds **131** and **132** was due to the absence of quinoline acetohydrazone moiety in final thiadiazoles [87].

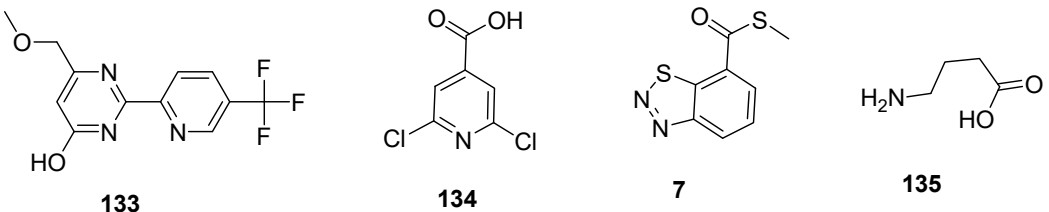

**131**          **132**

**Figure 23.** The most active antiamoebic 1,2,3-thiadiazoles **131** and **132**.

*3.5. Plant Activator Agents*

Plant activators are synthetic or natural chemicals entities that protect plants from different pathogens. Among them, pyrimidin-type plant activator 2 (PPA2 **133**; Figure 24) significantly enhanced plant defense system against bacterial infections unlikely traditional commercially available plant activators (INA **134**, BABA **135**, BHT **7**; Figure 24). Plant activators not only showed their therapeutic potential against wide variety of pathogens but also increased growth of plants, rate of photosynthesis and overall yield of crops [144–148].

**133**          **134**          **7**          **135**

**Figure 24.** Structures of commercially available plant activators **133**–**135**.

Du et al. reported a synthetic pathway for the preparation of carboxylate derivatives of thiadiazoles as potential plant activators against seven plant diseases [90]. The 1,2,3-thiadiazole derivatives **136** and **137** (Figure 25) proved to be good plant activators and displayed excellent inhibition activities against diseases such as *M. melonis*, (90% for **136** and 69% for **137**), *C. cassiicola* (77% for **136** and 52% for **137**), *P. syringae* pv. *Lachrymans* (42% for **136** and 42% for **137**) and *P. infestans* (81% for **136** and 67% for **137**) when results were compared with commercial BTH plant activator. The structural motifs **136** and **137** displayed significantly better therapeutic efficacy than BHT commercial drug, but the scaffold **136** had better plant activating potency than scaffold **137** [88].

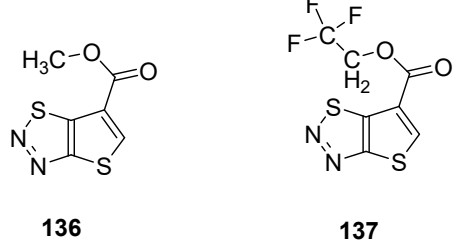

**136**          **137**

**Figure 25.** Potential new plant activator with thieno [2,3-*d*]-1,2,3-thiadiazoles **136** and **137**.

### 3.6. Fungicidal Agents

The chemical agents that have the potential to control and eradicate fungal infections are termed as fungicides or antifungal drugs [149]. The commercial antifungal agents are chlorothalonil (**65**) which is a non-systemic fungicide [150,151], fluconazole (**66**) [152] and ketoconazole (**67**) (Figure 26) [153].

**Figure 26.** Structures of commercial antifungal drugs **138**–**140**.

As the first example, it is worth bring up oxadiazoles containing thiadiazole derivatives [154]. In this group, two oxadiazole derivatives **141** and **142** (Figure 27) displayed the best antifungal activity against *Puccinia triticina* at a concentration of 500 μg/mL. Both structures showed a remarkable high growth inhibition activity (98 and 83%, respectively) which was slighter lower than the standard drug chlorothalonil with 100% fungal growth inhibition activity. The remarkable high and excellent antifungal activity of scaffold **141** could lead to the development of novel oxadiazole-based 1,2,3-thiadiazole fungicides [154].

**Figure 27.** Novel potent 1,2,3-thiadiazole fungicides **141** and **142**.

Another example of compounds with good antifungal activities are new structures designed by Sun et al., which are 1,2,4-triazole derivatives substituted with a 1,2,3-thiadiazole ring [82]. One of the synthesized compounds **143** (Figure 28), exhibited remarkably high fungicidal 93.19% against *Corynespora cassiicola,* while reference compounds iprodione, validamycin and topsin-M possessed 78.20, 53.52 and 78.75% antifungal activity, respectively. The authors declared that compound containing phenyl group on triazole ring **143** exhibited higher activity, however, the results were different in the case of *Pseudoperonospora cubensis* as compound **144** containing cyclopropyl group displayed higher activity than that of **143**. The most active compound with regard to *Pseudoperonospora cubensis* was **144**, which inhibited the growth of this pathogen in 81.62%. However, SAR analysis did not clear the results against *pseudomonas syringae* pv. *Lachrymans*. Both the scaffolds showed excellent and the highest antifungal potential than all three commercial reference drugs that will lead to the development of promising antifungal agents [82].

**Figure 28.** Antifungal activity of triazole moiety containing 1,2,3-thiadiazolew **143** and **144**.

Less antifungal activity was observed in the derivative **145** (Figure 29) reported by Tan et al. [155]. Its inhibition rates to *Phytophthora infestans* (Mont) de Bary, *Alternaria solani*, *Gibberella sanbinetti*, *Physalospora piricola* Nose, *Botrytis cinerea*, *Phytophthora capsici* Leonian, *Cercospora arachidicola*, *Rhizoctonia solanii* and *Fusarium oxysporum* reached 32.3, 17.4, 32.3, 25.8, 43.5%, 36.1, 15.8, 30.3 and 26.9% at 50 μg/mL respectively. The results indicated that the structural motif **145** proved to be broad-spectrum fungicide because it exhibited significant antifungal activity against different fungal strains [155].

**Figure 29.** Piperidine based 1,2,3-Thiadiazole **145** with moderate antifungal activity.

Another group reported the fungicidal properties of 1,2,3-thiadiazole derivatives against *Alternaria solani* (AS), *Botrytis cinerea* (BC), *Cercospora arachidicola* (CA), *Cercospora beticola* (CB), *Colletotrichum lagenarium* (CL), *Fusarium oxysporum* (FO), *Gibberella zeae* (GZ), *Macrophoma kuwatsukai* (MK), *Phytophthora infestans(Mont) de Bary* (PI), *Physalospora piricola* (PP), *Pellicularia sasakii* (Shirai) (PS), *Puccinia triticina Eriks* (PT) and *Rhizoctonia solani Kuhn* (RS) [156]. The most active occurred to be structure **146** (Figure 30) showed potent fungicidal inhibitory activity 84.8, 83.9, 83.3, 78.9, 75.0, 71.4, 70.7, 66.7, 64.7 and 63.6% against PS, PP, RS, PT, FO, CB, PI, AS, CL and CA, respectively. On the other hand, the propyl containing fused 1,2,4-triazolo[1,3,4]thiadiazole **147** (Figure 30) exhibited the best fungicidal inhibition activity 93.0% for FO, 84.9% for BC, 77.8% for PS, 75.8% for PI, 75.0% for GZ, 62.1% for CA, 50.0% for CL, 40.5% for PP, 34.4% for AS and 33.3% for CB. The results of the antifungal study showed that fused 1,2,4-triazolo[1,3,4]thiadiazole led to the development of wide-spectrum antifungal agents [156].

Wang et al. reported a synthetic approach to new series of organotin 4-methyl-1,2,3-thiadiazole-5-carboxylates and benzo[1,2,3]thiadiazole-7-carboxylates [157] and antifungal properties against *P. piricola* and *Gibberella zeae*. The triethyltin-based 1,2,3-thiadiazole carboxylate analogue **148** (Figure 31) displayed antifungal efficacy $EC_{50}$ = 0.12 μg/mL and 0.16 μg/mL against *P. piricola,* and *Gibberella zeae,* respectively. The overall results of the present study indicated the antifungal effectiveness of organotin-based benzo-1,2,3-thiadiazole or 1,2,3-thiadiazoles analogues proved to be broad-spectrum fungicides [157].

Antifungal properties have also been determined for carboxamide derivatives of thiadiazoles obtained by Zuo et al. utilized one-pot four-component Ugi reaction strategy under green synthetic conditions [158]. All the synthesized analogues were investigated against eleven fungal strains to develop novel fungicidal candidates. The 1,2,3-thiadiazole

containing carboxamide moiety **149** (Figure 32) displayed broad-spectrum fungicidal inhibition activities against CA 71%, AS 100%, GZ 62%, PP 85%, PG 14%, PS, 74%, CL 95%, PI 88% and RS 97%. Similarly good antifungal activity was observed for scaffold **150** that exhibited fungicidal inhibition against various fungal strains such as: CB (*Cercospora beticola*) 67%, CA 79%, AS 44%, GZ 34%, PP 53%, PG 57%, PS 16%, CL 43%, RS 51% and PI 57%. The scaffold **149** displayed maximum fungicidal potential 100, 97 and 95% against fungal strains AS, RS and CL, respectively, while the zero percent antifungal activity was shown against CB and FO fungal strains, just as scaffold **150** displayed the best antifungal potential 67% against CB fungal strains and 0% against FO [158].

**Figure 30.** 1,2,3-thiadiazoles with broad antifungal activity.

**Figure 31.** Antifungal benzo[1,2,3]thiadiazole-7-carboxylate derivative **148**.

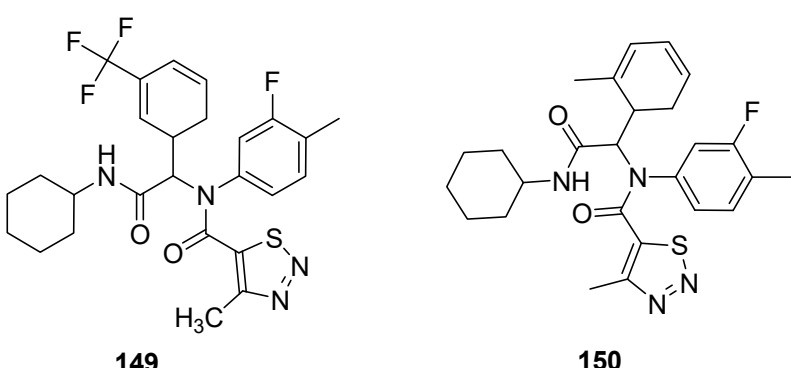

**Figure 32.** The most active 4-methyl-1,2,3-thiadiazole-5-carboxamides.

## 4. Conclusions

The review has uncovered 1,2,3-thiadiazole is a unique heterocyclic structural motif and privileged template in the field of medicinal chemistry, specifically its antiviral profile, to attract the researchers/scientists to plan and create novel, target-based and advanced 1,2,3-thiadiazole derivatives. A broad pharmaceutical profile and biological properties such as fungicidal, antiviral, insecticidal, amoebicidal, anticancer and plant activators,

etc. are shown along with a structure–activity relationship that will prompt potential pharmaceutical agents. In this review, we have tried to outline and summarize different synthetic approaches and strategies along with detailed modifications in substituents (structure–activity relationship). This review represents fruitful matrix that will help the researchers to develop lead compounds in different biological domains.

**Author Contributions:** Conceptualization: A.I.; resources: S.U., A.A., N.J., A.F.Z. and H.K.; writing—original draft preparation: A.I. and A.F.Z.; writing—review and editing: K.K.-M. and M.M. All authors have read and agreed to the published version of the manuscript.

**Funding:** This research received no external funding.

**Institutional Review Board Statement:** Not applicable.

**Informed Consent Statement:** Not applicable.

**Data Availability Statement:** Data sharing not applicable.

**Acknowledgments:** Authors are thankful to the Department of Chemistry, GCU Faisalabad for its literature survey facilities. There was no funding support.

**Conflicts of Interest:** The authors declare no conflict of interest.

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
