# Peer review of "Synthetic Transformations and Medicinal Significance of 1,2,3-Thiadiazoles Derivatives: An Update"

_applsci, doi:10.3390/app11125742_

Round 1
Reviewer 1 Report
This review could be accepted for publication after consideration of the following points:
Page 3, Scheme 2, a double bond is missing for 1,2,3-thiadiazole structure.
Page 5, Scheme 6, ethyl acetoacetate structure should be put above the arrow of the second reaction.
The links for reference 87 is wrong (same link for ref 88).
Page 7, Scheme 10, a double bond is missing for the final products.
Page 10, Scheme 19, “EtO2” should be “Et2O”
Author Response
Dear Reviewer,
Thank you for your opinion and suggestions regarding our manuscript.
Best regards
Mariusz Mojzych

Reviewer 2 Report
Irfan et. al., reported a review entitled: “Synthetic Transformations and Medicinal Significance of 1,2,3-Thiadiazoles derivatives: An update.” In this manuscript, authors were described synthetic approaches to 1,2,3-thiadiazole and its derivatives including its biological activities. Although, this manuscript is a collective information from a literature survey, this article is not very well written, and does not provided authors perspective on 1,2,3-thiadiazole. Entire article has grammatical errors. Also, schemes and figures, including its legends are not consistent. Some structures have distorted, with bold atoms, not have a correct bond angles. Moreover, this article does not have any new information from the reviews published in the past, including the review published by Irfan et. al., in 2019. This review does not have much reader scope and does not warrant for publication in this journal.
Author Response
Dear Reviewer,
Thank you for your feedback and suggestions regarding our manuscript. Graphical errors in diagrams and Figures have been corrected, as well as stylistic and grammatical errors in the text. We kindly ask you to re-check the manuscript and positively evaluate it.
Best Regards,
Mariusz Mojzych

Reviewer 3 Report
This manuscript focuses on the reviewing up-to-date knowledge on 1,2,3-thiadiazole hybrid structures, among others with important medical applications, such as: anti-fungal, antiviral, insecticidal, anti-amoebic, anti-cancer. Thus, the review presents different synthetic transformations of 1,2,3-thiadiazole scaffolds, their pharmaceutical profile and biological properties, also emphasizing SAR studies. The information presented was taken from a significant number of references and may be of interest for the scientific community.
Objections:
Figure 2 raises many objections:
The methyl group was represented by 3 distinct ways:
- “–“ for compunds 5, 6, 8 and 10;
- “Me” for 7, and
- “CH3” for 9, 11, 12, 13 and 14. It is necessary that the drawing of the chemical structures be uniform, at least within the same figure.
The compound 14 named “Cefazoline” is wrongly drawn, it is missing the “COOH” group.
Although in the text it is written that the compounds in Figure 2 were taken from the references [26, 72, 75, 76] certain compounds were not found in the mentioned references, e.g. compounds 5 and 14. Which compounds were taken from reference 76 for example?
In scheme 1 and Scheme 3 is missing HCl.
Scheme 5 does not specify who “R” is.
The structures of compound 64 from Figure 25 and 74 from Figure 30 are not drawn correctly.
For a correct drawing of the chemical structures check the IUPAC GRAPHICAL REPRESENTATION STANDARDS FOR CHEMICAL STRUCTURE DIAGRAMS.
Author Response
Dear Reviewer,
Thank you for your opinion and suggestions regarding our manuscript.
Best Regards,
Mariusz Mojzych

Round 2
Reviewer 2 Report
Although revised manuscripts looks good, still following few corrections are required to accept for publication.
- In Scheme 8, adjust the reaction arrow for 48 to 49.
- For compound 90, draw the alkyne in linear.
- Few structures have atoms in bold, should be unbold (Structures 127, 7, 141 and 142).
Author Response
The manuscript has been corrected according to the reviewer's comments.
